# Bridging the Gap Between Offline and Online Reinforcement Learning Evaluation Methodologies

**Shivakanth Sujit**                                   *shivakanth.sujit.1@ens.etsmtl.ca*
*ÉTS Montreal, Mila Quebec*

**Pedro H. M. Braga**                                          *phmb4@cin.ufpe.br*
*Universidade Federal de Pernambuco, Mila Quebec*

**Jorg Bornschein**                                    *bornschein@deepmind.com*
*DeepMind*

**Samira Ebrahimi Kahou**                       *samira.ebrahimi.kahou@gmail.com*
*ÉTS Montréal, Mila Quebec, CIFAR AI Chair*

**Reviewed on OpenReview:** *https://openreview.net/forum?id=J3veZdVpts*

## Abstract

Reinforcement learning (RL) has shown great promise with algorithms learning in environments with large state and action spaces purely from scalar reward signals. A crucial challenge for current deep RL algorithms is that they require a tremendous amount of environment interactions for learning. This can be infeasible in situations where such interactions are expensive, such as in robotics. Offline RL algorithms try to address this issue by bootstrapping the learning process from existing logged data without needing to interact with the environment from the very beginning. While online RL algorithms are typically evaluated as a function of the number of environment interactions, there isn't a single established protocol for evaluating offline RL methods. In this paper, we propose a sequential approach to evaluate offline RL algorithms as a function of the training set size and thus by their data efficiency. Sequential evaluation provides valuable insights into the data efficiency of the learning process and the robustness of algorithms to distribution changes in the dataset while also harmonizing the visualization of the offline and online learning phases. Our approach is generally applicable and easy to implement. We compare several existing offline RL algorithms using this approach and present insights from a variety of tasks and offline datasets. The code to reproduce our experiments is available at https://shivakanthsujit.github.io/seq_eval.

## 1 Introduction

Reinforcement learning (RL) has shown great progress in recent years with algorithms learning to play highly complex games with large state and action spaces such as DoTA2 ($\approx 10^4$ valid actions) and StarCraft purely from a scalar reward signal (Berner et al., 2019; Vinyals et al., 2019). However, each of these breakthroughs required a tremendous amount of environment interactions, sometimes upwards of 40 years of accumulated experience in the game (Schrittwieser et al., 2019; Silver et al., 2016; 2018). This can be infeasible for applications where such interactions are expensive, for example robotics.

Offline RL methods tackle this problem by leveraging previously collected data to bootstrap the learning process towards a good policy. These methods can obtain behaviors that maximize rewards obtained from the system conditioned on a fixed dataset of experience. The existence of logged data from industrial applications provides ample data to train agents in simulation till they achieve good performance and then

can be trained on real hardware. The downside of relying on offline data without any interactions with the system is that the behavior learned can be limited by the quality of data available (Levine et al., 2020).

Levine et al. (2020) point out that there is a lack of consensus in the offline RL community on evaluation protocols for these methods. The most widely used approach is to train for a fixed number of epochs on the offline dataset and report performance through the average return obtained over a number of episodes in the environment. However, this style of training does not provide insights into the data efficiency of offline RL algorithms. It also does not reveal how an algorithm reacts to changes in the dataset distribution, for example with better dataset-generating policies. In this paper, we propose to evaluate algorithms as a function of available data instead of just reporting final performance or plotting learning curves over a number of gradient steps. This approach allows us to study the sample efficiency and robustness of offline RL algorithms to distribution shifts. It also makes it easy to compare with online RL algorithms as well as intuitively study online fine-tuning performance. We call this approach of evaluation Sequential Evaluation (SeqEval) and present settings where SeqEval can be integrated into the evaluation protocol of offline RL algorithms. We also propose a style of model card (Mitchell et al., 2018) that can be designed for offline RL algorithms to provide relevant context to practitioners about their sample efficiency to aid in algorithm selection.

## 2 Background and Related Work

In Reinforcement Learning (RL) we are concerned with learning a parametric policy $\pi_\theta(a|s)$ that maximizes the reward it obtains from an environment. The environment is described as a Markov Decision Process (MDP) using $< \mathcal{S}, \mathcal{A}, \mathcal{R}, \mathcal{P} >$, where $\mathcal{S}$ represents the states of the environment, $\mathcal{A}$ represents the actions that can be taken, $\mathcal{R}$ is the set of rewards and $\mathcal{P}$ denotes the transition probability function $\mathcal{S} \times \mathcal{A} \to \mathcal{S}$. The objective is to learn an optimal policy, denoted by $\pi_\theta^*$, as $\pi_\theta^* = argmax_\theta \mathbb{E}\left[\sum_{t=0}^{\infty} \gamma^t r_t\right]$ where $\gamma \in [0, 1]$ is the discount factor and $r_t$ is the per timestep reward. The Q value of a policy for a given state-action pair is the expected discounted return obtained from taking action $a_t$ at state $s_t$ and following the policy $\pi$ afterwards. By estimating the Q value, we can implicitly represent a policy by always taking the action with the highest Q value. The Q value network is trained by minimizing the temporal difference (TD) error, defined as $(Q(s_t, a_t) - (r_t + \gamma \bar{Q}(s_{t+1}, a_{t+1}))$. $\bar{Q}$ is called the target Q network and is usually a delayed copy of the Q network or a exponentially moving average of the Q network.

**Offline RL.** The simplest form of offline RL is behavior cloning (BC) which trains an agent to mimic the behavior present in the dataset, using the dataset actions as labels for supervised learning. However, offline datasets might have insufficient coverage of the states and operating conditions the agent will be exposed to. Hence BC agents tend to be fragile and can often perform poorly when deployed in the online environment. Therefore, offline RL algorithms have to balance two competing objectives, learning to generalize from the given dataset to novel conditions without deviating too far from the state space coverage in the dataset. The former can be enabled by algorithms that learn to identify and mimic portions of optimal behavior from different episodes that are suboptimal overall, as pointed out in Chen et al. (2021). The latter issue is addressed by constraining the agent's policy around behavior seen in the dataset, for example, through enforcing divergence penalties on the policy distribution (Peng et al., 2019; Nair et al., 2020). Another way of penalizing out of dataset predictions is by using a regularizer on the Q value, for example Conservative Q Learning (CQL) (Kumar et al., 2020) prevents actions that have low support in the data distribution from having high Q values. Decision Transformer (DT) (Chen et al., 2021) does not learn a Q estimate and instead formulates the RL task as a sequence modelling task conditioned on the return-to-go from a given state. The agent is then queried for the maximum return and predicts actions to generate a sequence that produces the queried return. This is a very brief overview of offline RL methods, and we direct readers to Levine et al. (2020) for a broader overview of the field.

**Metrics and Objectives.** The paradigm of *empirical risk minimization* (ERM) (Vapnik, 1991) is the prevailing training and evaluation protocol, both for supervised and unsupervised Deep Learning (DL). At its core, ERM assumes a fixed, stationary distribution and that we are given a set of (i.i.d.) data points for training and validation. Beyond ERM, and especially to accommodate non-stationary situations,

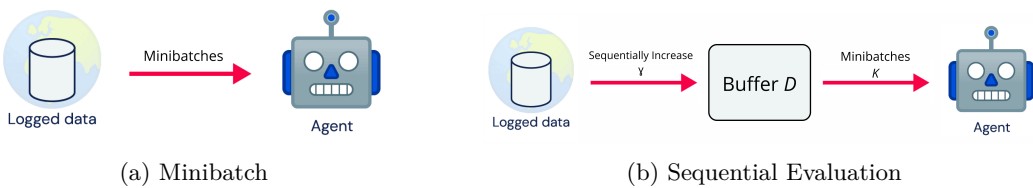

(a) Minibatch                                  (b) Sequential Evaluation

Figure 1: Comparison of traditional training schemes with Sequential Evaluation

---

**Algorithm 1** Algorithm for Sequential Evaluation in the offline setting.

---

1: **Input:** Algorithm $A$, Offline data $\mathcal{D} = \{s_t, a_t, r_t, s_{t+1}\}_{t=1}^{T}$, increment-size $\gamma$, gradient steps per increment $K$, evaluation frequency $f_e$
2: Replay-buffer $\mathcal{B} \leftarrow \{s_t, a_t, r_t, s_{t+1}\}_{t=1}^{T_0}$
3: $t \leftarrow T_0$
4: **while** $t < T$ **do**
5:     Update replay-buffer $\mathcal{B} \leftarrow \mathcal{B} \cup \{s_t, a_t, r_t, s_{t+1}\}_t^{t+\gamma}$
6:     Sample a training batch, ensure new data is included: batch $\sim \mathcal{B}$
7:     Perform training step with $A$ on batch.
8:     $t \leftarrow t + \gamma$
9:     **for** $j = 1, \cdots, K$ **do**
10:        Sample a training batch $\sim \mathcal{B}$
11:        Perform training step with $A$ on batch.
12:     **end for**
13:     **if** t % $f_e = 0$ **then**
14:        Evaluate $A$ in the environment and log performance.
15:     **end if**
16: **end while**

---

different fields have converged to alternative evaluation metrics: In online learning, bandit research, and sequential decision making in general, the (cumulative) reward or the *regret* are of central interest. The regret is the cumulative loss accrued by an agent relative to an optimal agent when sequentially making decisions. For learning in situations where a single, potentially non-stationary sequence of observations is given, Minimum Description Length (MDL) (Rissanen, 1984) provides a theoretically sound approach to model evaluation. Multiple, subtly different formulations of the description length are in use, however, they are all closely related and asymptotically equivalent to *prequential MDL*, which is the cumulative log loss when sequentially predicting the next observation given all previous ones (Rissanen, 1987; Poland & Hutter, 2005). Similar approaches have been studied and are called the *prequential approach* (Dawid & Vovk, 1999) or simply *forward validation*. A common theme behind these metrics is that they consider the agents' ability to perform well, not only in the big-data regime, but also its generalization performance at the beginning, when only a few observations are available for learning. The MDL literature provides arguments and proofs why those models, that perform well in the small-data regime without sacrificing their big-data performance, are expected to generalize better to future data (Rathmanner & Hutter, 2011).

With these two aspects in mind, that a) RL deals with inherently non-stationary data, and b) that sample efficiency is a theoretically and practically desirable property, we propose to evaluate offline RL approaches by their data efficiency.

## 3   Sequential Evaluation of Offline RL Algorithms

As mentioned above, one approach for offline RL evaluation is to perform multiple epochs of training over the dataset. We contend that there are a few issues with this approach. Firstly, this approach does not provide much information about the sample efficiency of the algorithm since it is trained on all data at every epoch. This means that practitioners do not see how the algorithm can scale with the dataset size,

or if it can achieve good performance even with small amounts of logged data. Furthermore, there can be distribution changes in the quality of the policy in the dataset, and evaluating as a function of epochs hides how algorithms react to these changes. Finally, there is a disconnection in the evaluation strategies of online and offline RL algorithms, which can make it difficult to compare algorithms realistically.

Instead of treating the dataset as a fixed entity, we propose that the portion of the dataset available to the agent changes over time and that the agent's performance is evaluated as a function of the available data. We term this as sequential evaluation of offline algorithms (SeqEval). This can be implemented by reusing any of the prevalent replay-buffer-based training schemes from online deep RL. Instead of extending the replay-buffer with sampled trajectories from the currently learned policy, we insert prerecorded offline RL data. That is, the offline dataset replaces the environment as the data generating function. We alternate between adding new samples to the buffer and performing gradient updates using mini-batches sampled from the buffer. The number of samples added to the buffer at a time is denoted by $\gamma$ and the number of gradient steps performed between each addition to the buffer is denoted by $K$. A concrete implementation of the approach is outlined in Alg. 1 and it is illustrated in Fig. 1.

This approach of evaluation addresses several of the issues with epoch-style training. By varying $\gamma$ and $K$ we can get information about the scaling performance of an algorithm with respect to dataset size, which tells us if data is the bottleneck for further improvements, and how quickly the algorithm can learn with limited data. We can visualize how the algorithm behaves with shifts in dataset quality directly from the performance curves. $\gamma$ and $K$ together affect the rate at which an offline RL algorithm is given access to the data and hence can be studied together depending on the needs of the practitioner. Online RL methods are evaluated as a function of the number of environment steps, which is a measure of the amount of data available to the agent. In the offline RL setting this would translate to evaluating the algorithm in terms of the amount of data it has access to in the replay buffer. We can also seamlessly evaluate the performance of the algorithm in online fine-tuning by adding samples from the environment once the entire offline dataset is added to the replay buffer. A benefit of the sequential approach is that it does not require a complete overhaul in codebases that follow existing training paradigms. For the baselines that we present in this paper, we were able to use the sequential evaluation approach with less than 10 lines of changes to the original codebases.

The ratio $\frac{K}{\gamma}$ is the replay ratio (RR), a hyperparameter commonly found in RL algorithms which use replay-buffers. The RR determines how many gradient steps are performed for each added data point. Increasing RR allows some algorithms to extract more utility from the data it has received, potentially improving sample efficiency. By varying RR we can identify whether computation is the bottleneck for improving performance, or the amount of available data.

**Implementation Details** To ensure that the algorithm sees each data point in the dataset at least once, when a new batch of data is added to the buffer, the algorithm is trained on that sample of data once before $K$ mini-batches are sampled from the buffer for training. If the batch added is smaller than the mini-batch size, then it can be made part of the next mini-batch that is trained on. In practice, we found that setting $\gamma$ and $K$ to 1 worked well in all datasets tested. This means that the x-axis of all plots directly corresponds to the number of samples available for training and the number of gradient updates performed. Additionally, we study other values of the RR and report these results in Appendix C. We observe that a RR of 1 generally performs well across datasets and higher RRs lead to overfitting and worse performance. The changes made to the codebase of each algorithm are as follows: Each codebase had a notion of a replay buffer that was being sampled, and the only addition required here was a counter that kept track of up to which index in the buffer data points could be sampled from to create mini-batches. The counter was initialized to $T_0 = 5000$ so that there were some samples in the buffer at the start of training. The second change that needed to be made was changing the outer loop of training from epochs to the number of gradient updates and incrementing the buffer counter by $\gamma$ every $K$ update. This way, the amount of data the algorithm was trained on sequentially increased to the full dataset over the course of training. Finally, we shuffle the order of the dataset on each seed so that the training curves are not specfic to only one particular ordering of the dataset. From our experiments we see that the dataset order does not have a significant impact on the performance of the algorithm.

## 4 Experiments

### 4.1 Baselines and Benchmarks

We evaluate several existing offline RL algorithms using the sequential approach, namely Implicit Q Learning (IQL) (Kostrikov et al., 2022), CQL (Kumar et al., 2020), TD3+BC (Fujimoto & Gu, 2021), AWAC (Nair et al., 2020), BCQ (Fujimoto et al., 2019), Decision Transformer (DT) (Chen et al., 2021) and Behavior Cloning (BC). These algorithms were evaluated on the D4RL benchmark (Fu et al., 2020), which consists of three environments: Halfcheetah-v2, Walker2d-v2 and Hopper-v2. For each environment, we evaluate four versions of the offline dataset: random, medium, medium-expert, and medium-replay. Random consists of 1M data points collected using a random policy. Medium contains 1M data points from a policy that was trained for one-third of the time needed for an expert policy, while medium-replay is the replay buffer that was used to train the policy. Medium-expert consists of a mix of 1M samples from the medium policy and 1M samples from the expert policy. These versions of the dataset are used to evaluate the performance of offline agents across a spectrum of dataset quality.

We also created a dataset from the DeepMind Control Suite (DMC) (Tassa et al., 2018) environments following the same procedure as outlined by the authors of D4RL. We chose the DMC environments because of their high dimensional state space and action space. Specifically we chose cheetah run, walker run and finger turn hard and trained a Soft Actor Critic (SAC) (Haarnoja et al., 2018) agent on these environments for 1M steps. A medium and expert version of the dataset was created for each of the three environments using policies obtained after 500K and 1M steps respectively.

Finally, to study algorithms in visual offline RL domains, we used the v-d4rl benchmark (Lu et al., 2023) which follows the philosophy of D4RL and creates datasets of images from the DMC Suite with varying difficulties. We use the cheetah run, humanoid walk and walker walk domains and the expert, medium and random versions of these domains. Lu et al. (2023) open sourced implementations of DrQ (Kostrikov et al., 2020) variants along with the dataset and we compare these algorithms using SeqEval.

### 4.2 Experimental Settings

We study varied ways in which SeqEval can be integrated into offline RL training. The first setting is the "Standard" setting in which samples are added periodically to the buffer during training. The results of the Standard setting on a subset of datasets are given in Fig. 2 and the complete set of datasets, along with an experiment comparing curves with larger $K$ are available in the appendix. Secondly, to highlight how SeqEval can visualize how the algorithm reacts to changing dataset quality during training, we create a "Distribution Shifts" setting where a mixed version of each environment is created. In this dataset, the first 33% of data comes from the random dataset, the next 33% from the medium dataset and the final 33% from the expert dataset. From the performance curves given in Fig. 6 we can see how each algorithm adapts to changes in the dataset distribution. We then show how SeqEval can be utilised in multi task offline RL in a similar fashion. Instead of providing the entire dataset of a new task to the agent at once, we periodically add new samples of the task to the agent over time. This lets us study the amount of samples required for the agent to transfer to the new task. Finally we also show how SeqEval supports seamless integration of online fine-tuning experiments into performance curves. In this setting, once the entire offline dataset is added to the replay buffer, the agent is allowed to interact with the online simulator for a fixed number of steps (500k steps in our experiments). Since the curves are a function of data samples, we can continue evaluating performance as before. The results on a subset of datasets are given in Fig. 7, and curves for all datasets are available in the appendix.

For each dataset, we train algorithms following Alg. 1, initializing the replay buffer with 5000 data points at the start of training. We set $\gamma$ and $K$ each to 1, that is, there is one gradient update performed on a batch of data sampled from the environment every time a data point is added to the buffer. The x-axis in the performance curves represents the amount of data in the replay buffer. In each plot, we also include the performance of the policy that generated the dataset as a baseline, which provides context for how much information each algorithm was able to extract from the dataset. This baseline is given as a horizontal dotted line.

### 4.3 Model Cards

Since SeqEval evaluates algorithms as a function of data, it opens up the possibility of quantitatively analysing how algorithms scale with data. We propose a few evaluation criteria in addition to the training curves. The first one is Perf@50%, which measures the aggregate performance of algorithms when half the dataset is available to it for training. To further analyze the role of data scaling, we studied two measures, Perf@50% normalised with respect to performance obtained with the whole dataset (Perf@100%) and the difference between Perf@100% and Perf@50%. These three metrics give end users valuable insights that can aid in offline algorithm selection. They can be presented as part of an overall model card (Mitchell et al., 2018) available to the user.

### 4.4 Results

**Standard** We present the results of the Standard setting in Figs. 2, 3 and 4. One striking observation from the learning curves in Figs. 2 and 4 is how quickly the algorithms converge to a given performance level and then stagnate. This is most evident in the medium version of each environment. With less then 300K data points in the buffer, each algorithm stagnates and does not improve in performance even after another 500K points are added. That is, there are diminishing returns from adding data beyond 500K points to the buffer. This highlights that most of the tested algorithms are not very data-hungry. That is, they do not require a large data store to reach good performance, which is beneficial when they need to be employed in practical applications. The experiment highlights that the chosen datasets might lack diversity in collected experience since most algorithms appear to need only a fraction of it to attain good performance. This phenomenon is made more stark in Fig. 5. Most algorithms reach within 90% of final performance with just 50% of the data. In other words, doubling the data in the D4RL benchmarks provides an aggregated uplift of 10% at best. This indicates that more effort could be put into designing a harder benchmark. The effect is less pronounced in the DMC datasets, where the algorithms continue learning till about 50% of the dataset is added and then learning begins to stagnate.

In a majority of the datasets tested on D4RL, CQL reaches better performance than other methods earlier in training and consistently stays at that level as more data is added. TD3+BC exhibits an initial steep rise in performance but levels out at a lower score overall, or in the case of Halfcheetah-medium-expert, degrades in performance as training progresses. In the visual domain, there is a similar trend with steep increases in performance for the first 100K points and much more modest gains as the dataset size doubles. These experiments highlight the need to further study the effect of dataset sizes in offline RL benchmarks since current algorithms appear to learn very quickly from limited data.

**Distribution Shifts** The learning curves for the "mixed" dataset in the D4RL benchmark and DMC are given in Fig. 6. In the mixed dataset experiment, most of the algorithms do adapt to the new data continually improving each time there is a shift in the dataset quality. However rarely are they able to do so in every single environment. While CQL adapts quickly in both HalfCheetah and Walker2d, it fails to learn at all in Hopper. TD3+BC outperforms all other methods in HalfCheetah, but is the worse performing in Walker2d. DT is able to consistently adapt to new data in each environment. A surprising result is how well BC performs in each environment, with BC nearly being the second best performing algorithm in all environments. We hypothesise that since there is good coverage of the state spaces and optimal behavior in the dataset, behavior cloning is able to learn effectively. It does not have to "stitch" behavior from different episodes and instead just need to replicate the behavior in the dataset. This is in line with the findings of Chen et al. (2021) which found that 10%BC, which trained BC on only the top 10% percentile of episodes (based on episodic return), was competitive with specialised offline algorithms on the D4RL benchmark.

**Online Finetuning** In online fine-tuning, shown in Fig. 7 and Table 3, CQL reaches higher scores compared to the other algorithms showing the versatility of CQL in handling both offline datasets and online interactions with the system. While in the random dataset there is a jump in performance after online finetuning, there aren't large gains in other datasets. This is evident from Fig. 5c, where we see that in general there are modest gains from online finetuning in these algorithms.

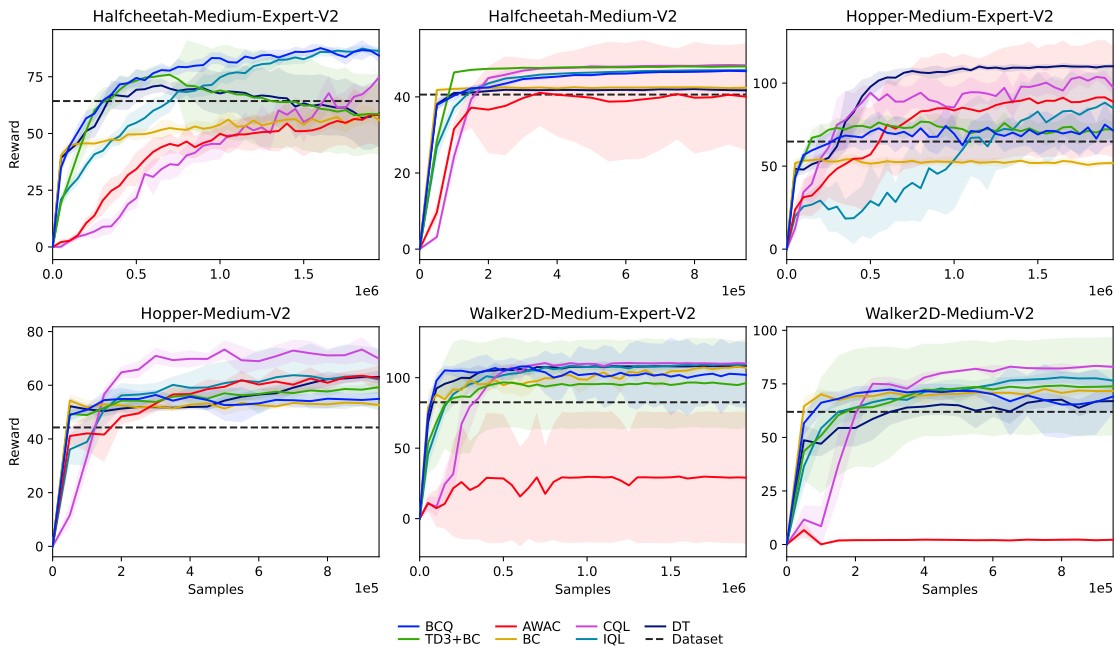

Figure 2: Performance curves on the D4RL benchmark as a function of data points seen. Shaded regions represent standard deviation across 5 seeds.

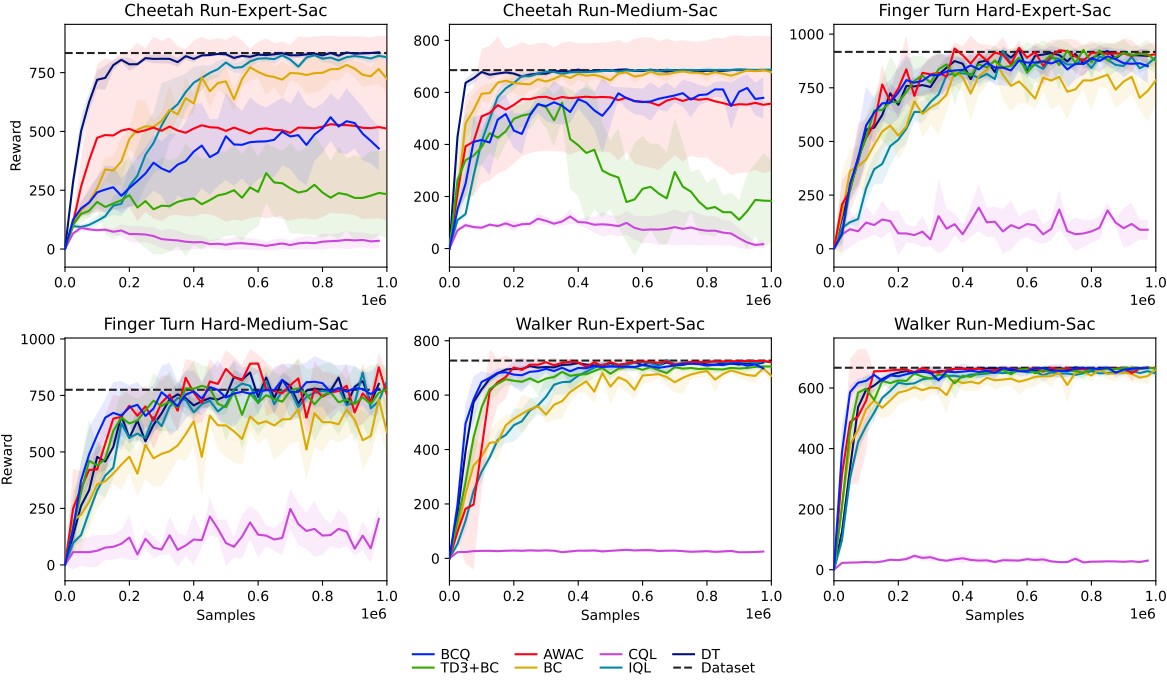

Figure 3: Performance curves on the DMC Dataset as a function of data points seen. Shaded regions represent standard deviation across 5 seeds.

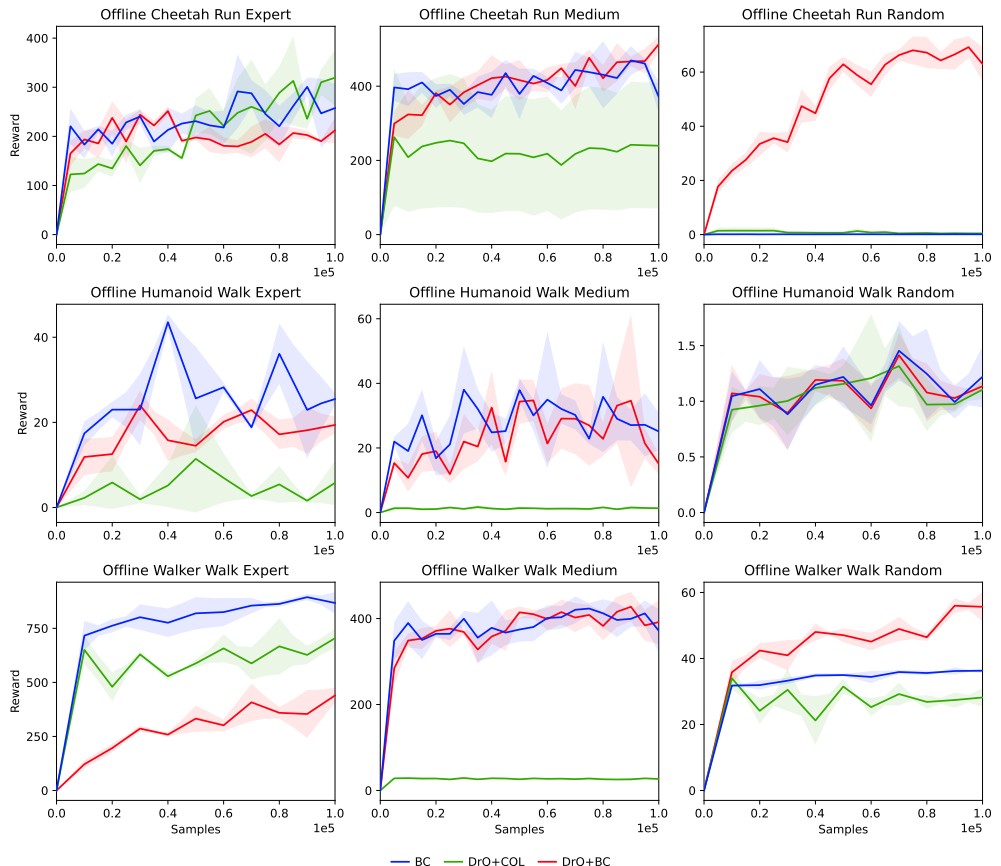

Figure 4: Performance curves on the v-d4rl benchmark as a function of data points seen. Shaded regions represent standard deviation across 5 seeds.

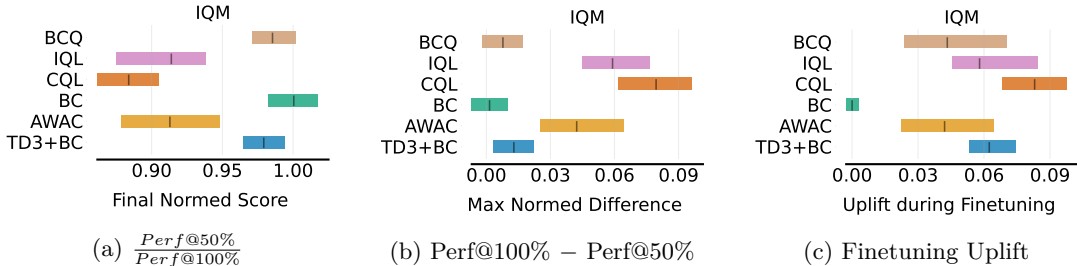

Figure 5: Analysis of data scaling in the D4RL benchmark. Perf@X% refers to performance when the algorithm has seen $X\%$ of the dataset. a) and b): Standard Setting c) Online Finetuning setting.

## 4.5 Discussion of Results

A surprising insight from our experiments was how the algorithms studied learnt with a fraction of the dataset size and learning stagnated after 20-30% of the dataset was added. Further addition of data did not lead to improvements in performance. This observation was consistent across environments and algorithms. We can view these results in two ways.

One, the current benchmarks/datasets for offline RL algorithms need to be re-evaluated. If an algorithm can perform similarly with 20% of the data or 100% of the data, then these datasets might not be sufficiently

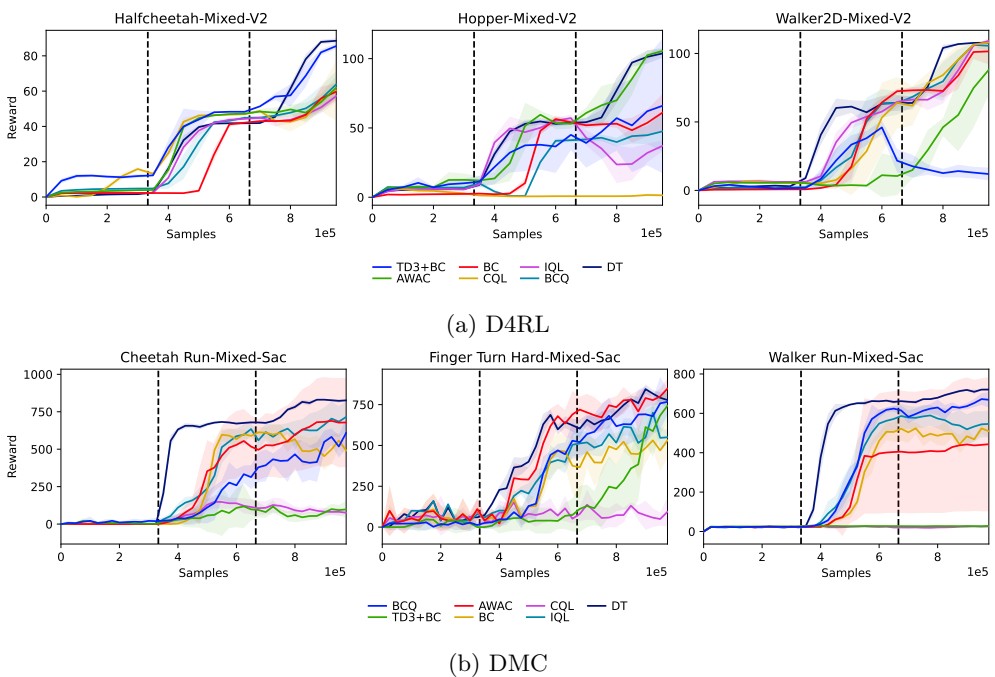

(a) D4RL

(b) DMC

Figure 6: Performance curves for Mixed datasets with varying dataset quality. Dotted line indicates where there is a change in the dataset generating policy distribution. Shaded regions represent standard deviation across 5 seeds.

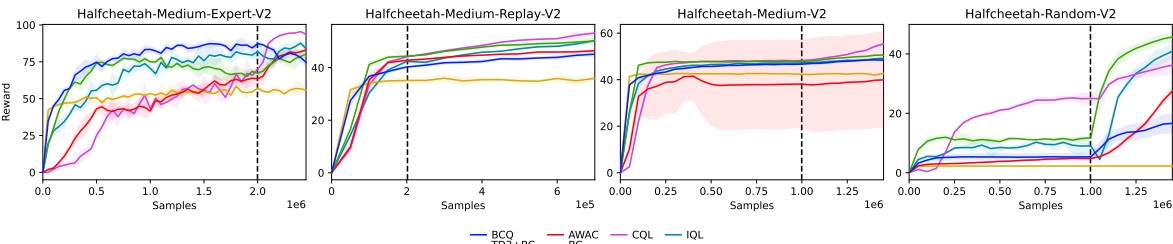

Figure 7: Performance curves for online fine-tuning. Each algorithm is given 500k steps in the simulator after sequential evaluation of the offline dataset. Dotted line indicates where online fine-tuning begins. Shaded regions represent standard deviation across 3 seeds.

difficult to properly evaluate algorithms. We believe that effort should be invested in designing harder benchmarks for this domain.

The second takeaway is that the algorithms tested were surprisingly data efficient. The model cards in Figs. 5a and 5b show that most algorithms reach around 90% of final performance with less than 50% of the dataset. Presenting such summary statistics can make it easy for practitioners to 1) perform algorithm selection, and 2) evaluate the benefits of additional data collection. The latter can be useful in practical domains where data collection is expensive and time consuming and the model cards can provide insights into the trade-off between better algorithmic performance and increased cost of data collection.

### 4.6 Comparison with Mini Batch Style Training

We compare how algorithms perform when given access to the same amount of compute in the SeqEval setting and the mini batch setting. The results are given in Fig. 8. In the mini batch style training, the algorithm is given access to the entire dataset at all times during training. The x axis in Fig. 8 is the

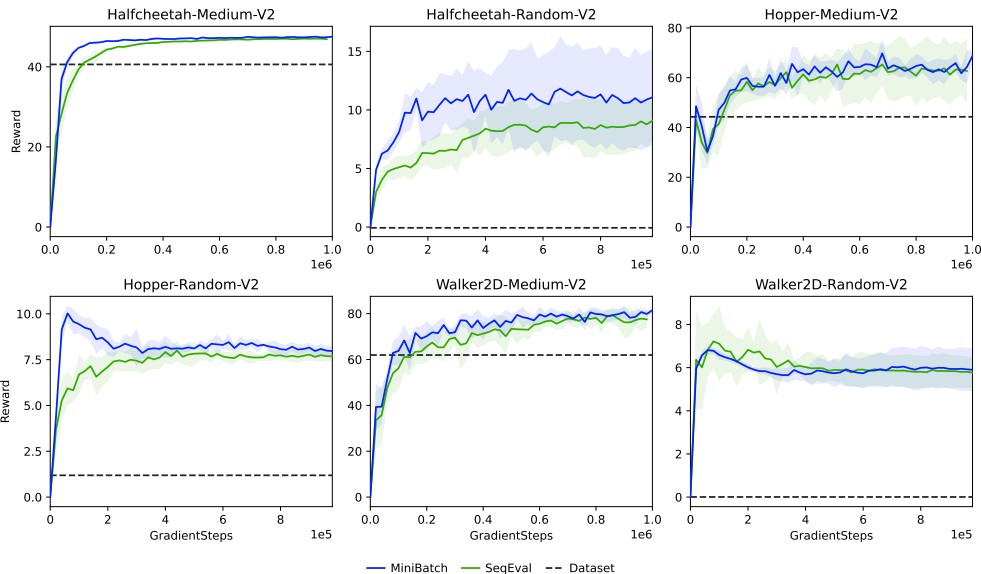

Figure 8: Comparison of SeqEval training and mini batch style training.

number of gradient steps, while in the previous figures for SeqEval the x axis represents the number of data points available for training. However when $\gamma = 1$ and $K = 1$, the x axis can also represent the number of gradient steps, hence we plot the SeqEval and mini batch training curves together. So while the algorithm has performed a similar amount of computation when comparing the curves at a given gradient step, the data they have access to at these points can be very different. As a result, SeqEval would exhibit a slower rise in performance per gradient step compared to mini batch, but that is because it simply does not have access to the same amount of data as mini batch style training. We do see that SeqEval closes the gap towards the end of training.

### 4.7 Additional Metrics for Evaluation

While we have used the evaluation episodic return as the performance metric in our experiments, it is not necessary to do so. For example, if there are environments where a simulator is not available for periodic cheap evaluation, off-policy evaluation metrics (Thomas et al., 2015; Wang et al., 2020; Munos & Szepesvári, 2008) can be used. We provide an example of this in Fig. 9 where instead of monitoring the episodic return during training, we utilize Fitted Q Evaluation (FQE) score (Wang et al., 2020). We followed the same implementation as given in Wang et al. (2020), training the FQE estimator from scratch each time during an evaluation phase. The FQE estimator was given access to the entire dataset during its training, while IQL still followed the SeqEval style of training. The exact FQE metric used was $\mathbb{E}[\hat{Q}(s_0, \pi(s_0))]$, i.e. the predicted Q value of the start state distribution.

We would like to emphasize that SeqEval is independent of the evaluation metric used during training. SeqEval is a style of training and can be used with different evaluation metrics depending on the constraints, for example simulator return or off-policy evaluation metrics such as FQE scores. The experiment above is a concrete example of how SeqEval can be used in situations where there is limited access to a simulator to evaluate performance.

## 5 Conclusion

In this paper, we propose a sequential style of evaluation for offline RL methods so that algorithms are evaluated as a function of data rather than compute or gradient steps. In this style of evaluation, data is added sequentially to a replay buffer over time, and mini-batches are sampled from this buffer for training.

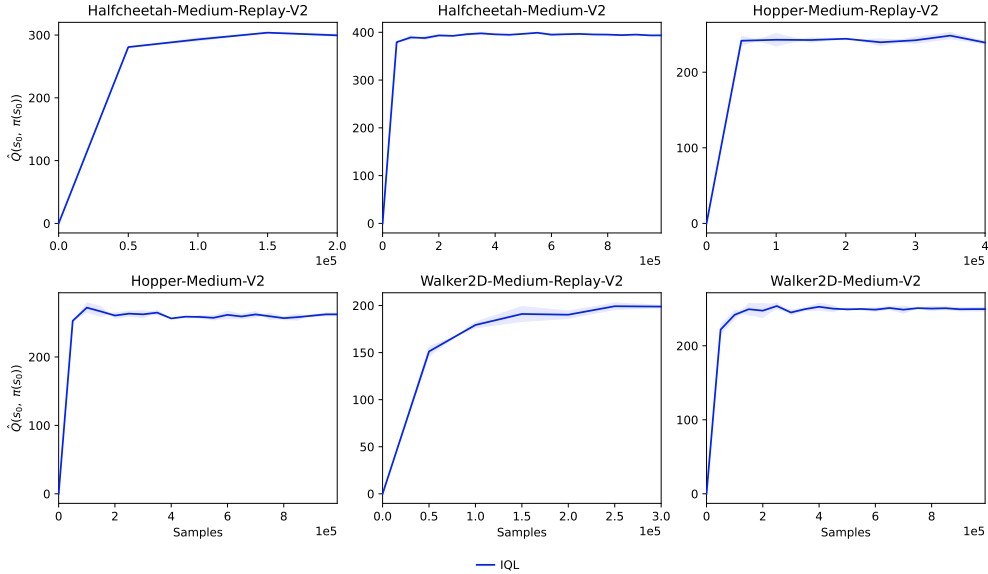

Figure 9: Performance of IQL in the SeqEval setting using FQE as the evaluation metric.

This is analogous to online training to deep RL and allows us to measure the data scaling and robustness of offline algorithms simultaneously from the training curves. We compared several existing offline methods using sequential evaluation and showed how their training curves allow for algorithm selection depending on data efficiency or performance.

We believe that sequential evaluation holds promise as an established method of evaluation for the offline RL community. Future work in this domain could explore the effect of $\gamma$ and $K$ on algorithms and their ramifications on sample efficiency.

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

## A  Description of datasets

The D4RL locomotion benchmark consists of three environments with varying data quality. The size of each version and the performance of the policy that generated it is given in Table 1.

Table 1: Dataset sizes in D4RL

| Dataset | Size | Average score in Dataset |
|---|---|---|
| halfcheetah-random-v2 | 999000 | -0.07 |
| halfcheetah-medium-v2 | 999000 | 40.60 |
| halfcheetah-medium-replay-v2 | 201798 | 27.03 |
| halfcheetah-medium-expert-v2 | 1998000 | 64.29 |
| halfcheetah-expert-v2 | 999000 | 87.91 |
| walker2d-random-v2 | 999999 | 0.01 |
| walker2d-medium-v2 | 999322 | 61.94 |
| walker2d-medium-replay-v2 | 301698 | 14.81 |
| walker2d-medium-expert-v2 | 1998318 | 82.55 |
| walker2d-expert-v2 | 999000 | 106.90 |
| hopper-random-v2 | 999999 | 1.19 |
| hopper-medium-v2 | 999998 | 44.28 |
| hopper-medium-replay-v2 | 401598 | 14.97 |
| hopper-medium-expert-v2 | 1998966 | 64.78 |
| hopper-expert-v2 | 999061 | 108.24 |

## B  Performance on D4RL Benchmark

We present complete training curves on all twelve datasets that were used in Fig. 11 and final performance in Table 2. In addition to the curves, we compare the algorithms at the end of training with scores aggregated across environments. This is done using the rliable (Agarwal et al., 2021) library to plot interval estimates of normalized performance measures such as median, mean, interquartile mean (IQM) and optimality gap. The optimality gap is a measure of how far an algorithm is from optimal performance aggregated across environments. So, lower values are better. The scores in each dataset are normalized with respect to the maximum score, which is 100. These results are given in Fig. 13. In both the performance curves and aggregated scores we can see that CQL outperforms other tested methods by a clear margin. A curious phenomenon observed was that AWAC (Nair et al., 2020) is unable to learn at all in the Walker2d environment with either the medium-expert or the medium version achieving very low rewards. As a sanity check we set $\gamma$ to the size of the dataset and reran experiments and the method achieved results similar to those reported in the paper, indicating it was not an implementation problem. This is surprising since AWAC is proposed as an algorithm that can work for online fine-tuning following offline pretraining, but among all the methods tested, it had drastic changes in performance when using sequential evaluation. The final performance in the online fine-tuning task and the mixed version of the environment is also given in Tables 3 and 4. The Perf@50% and Perf@100% results are given in Fig. 10

Moreover, Fig. 14 show how the algorithms performed when $K$ was set to 2. This experiment studied if we had not trained the methods for enough gradient steps and if additional performance could be extracted from the data. However, performance remained essentially the same or even degraded in some instances, showing that this was not the case.

## C  Effect of Replay Ratio in SeqEval

The experiments in the paper studied a replay ratio (RR) of 1. The results are in Figs. 15 and 16. We can directly compare the training and evaluation curves for different RRs and conclude whether compute (gradient-steps) or training data are the limiting factor. In our main experiments we study the setting where

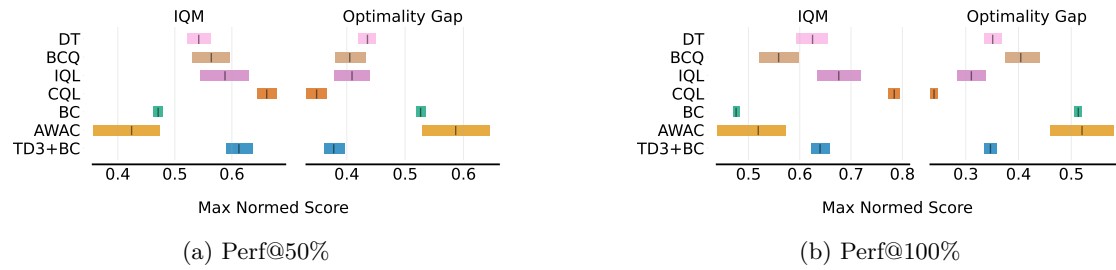

Figure 10: Comparison of performance when a) half the dataset is available b) full dataset is available

Table 2: Performance of each algorithm on the D4RL Benchmark

| Dataset | BCQ | TD3+BC | AWAC | BC | CQL | DT | IQL |
|---|---|---|---|---|---|---|---|
| halfcheetah-medium-expert-v2 | 92.25 | 6.83 | 59.04 | 63.44 | 83.45 | 71.6 | 86.0 |
| halfcheetah-medium-replay-v2 | 37.54 | 44.26 | 43.37 | 36.3 | 43.54 | 30.91 | 41.97 |
| halfcheetah-medium-v2 | 45.47 | 48.45 | 46.58 | 43.12 | 49.13 | 42.65 | 46.96 |
| halfcheetah-random-v2 | 8.16 | 10.09 | 2.26 | 2.26 | 24.43 | 2.12 | 8.41 |
| hopper-medium-expert-v2 | 108.8 | 69.92 | 111.88 | 44.53 | 111.0 | 111.38 | 49.18 |
| hopper-medium-replay-v2 | 23.9 | 63.04 | 65.47 | 14.02 | 88.51 | 74.94 | 69.09 |
| hopper-medium-v2 | 53.39 | 49.77 | 52.8 | 55.84 | 70.53 | 62.06 | 67.49 |
| hopper-random-v2 | 7.16 | 8.13 | 9.18 | 2.26 | 6.2 | 7.41 | 7.63 |
| walker2d-medium-expert-v2 | 110.36 | 108.33 | 1.98 | 107.58 | 109.98 | 108.23 | 98.51 |
| walker2d-medium-replay-v2 | 59.94 | 76.8 | 82.54 | 24.63 | 73.31 | 55.04 | 63.12 |
| walker2d-medium-v2 | 76.85 | 83.4 | 1.76 | 78.78 | 83.16 | 65.79 | 80.85 |
| walker2d-random-v2 | 0.11 | 0.49 | 3.48 | 0.63 | -0.12 | 2.48 | 7.82 |

Table 3: Performance of each algorithm in the online fine-tuning task on the D4RL Benchmark

| Dataset | BCQ | TD3+BC | AWAC | BC | CQL | IQL |
|---|---|---|---|---|---|---|
| finetune-halfcheetah-medium-expert-v2 | 88.02 | 74.36 | 83.6 | 61.77 | 96.72 | 87.74 |
| finetune-halfcheetah-medium-v2 | 47.86 | 51.9 | 52.89 | 42.86 | 55.29 | 49.24 |
| finetune-halfcheetah-random-v2 | 22.29 | 48.53 | 30.67 | 2.26 | 34.17 | 50.87 |
| finetune-hopper-medium-expert-v2 | 43.19 | 112.71 | 111.9 | 48.01 | 100.25 | 110.66 |
| finetune-hopper-medium-v2 | 52.35 | 63.03 | 45.45 | 58.92 | 94.31 | 73.59 |
| finetune-hopper-random-v2 | 14.31 | 9.85 | 8.78 | 2.47 | 4.56 | 12.05 |
| finetune-walker2d-medium-expert-v2 | 108.28 | 107.77 | 1.41 | 108.35 | 110.73 | 110.81 |
| finetune-walker2d-medium-v2 | 71.26 | 85.01 | 21.32 | 66.47 | 83.7 | 84.24 |
| finetune-walker2d-random-v2 | 1.42 | 8.6 | 1.67 | 0.5 | 0.33 | 10.56 |

Table 4: Performance of each algorithm in the mixed version of the D4RL Benchmark

| Dataset | BCQ | TD3+BC | AWAC | BC | CQL | DT | IQL |
|---|---|---|---|---|---|---|---|
| halfcheetah-mixed-v2 | 66.98 | 80.65 | 29.55 | 57.38 | 93.47 | 81.51 | 68.04 |
| hopper-mixed-v2 | 55.29 | 112.42 | 110.36 | 86.67 | 0.75 | 111.74 | 51.37 |
| walker2d-mixed-v2 | 102.77 | 8.15 | 98.36 | 108.79 | 109.71 | 107.63 | 109.83 |

RR = 1. We utilized this setting since we observed that increasing the compute budget did not improve performance. Lower RRs do not extract as much information as possible from the dataset and do worse. These curves provide valuable insights into the combined data and compute scaling of an algorithm.

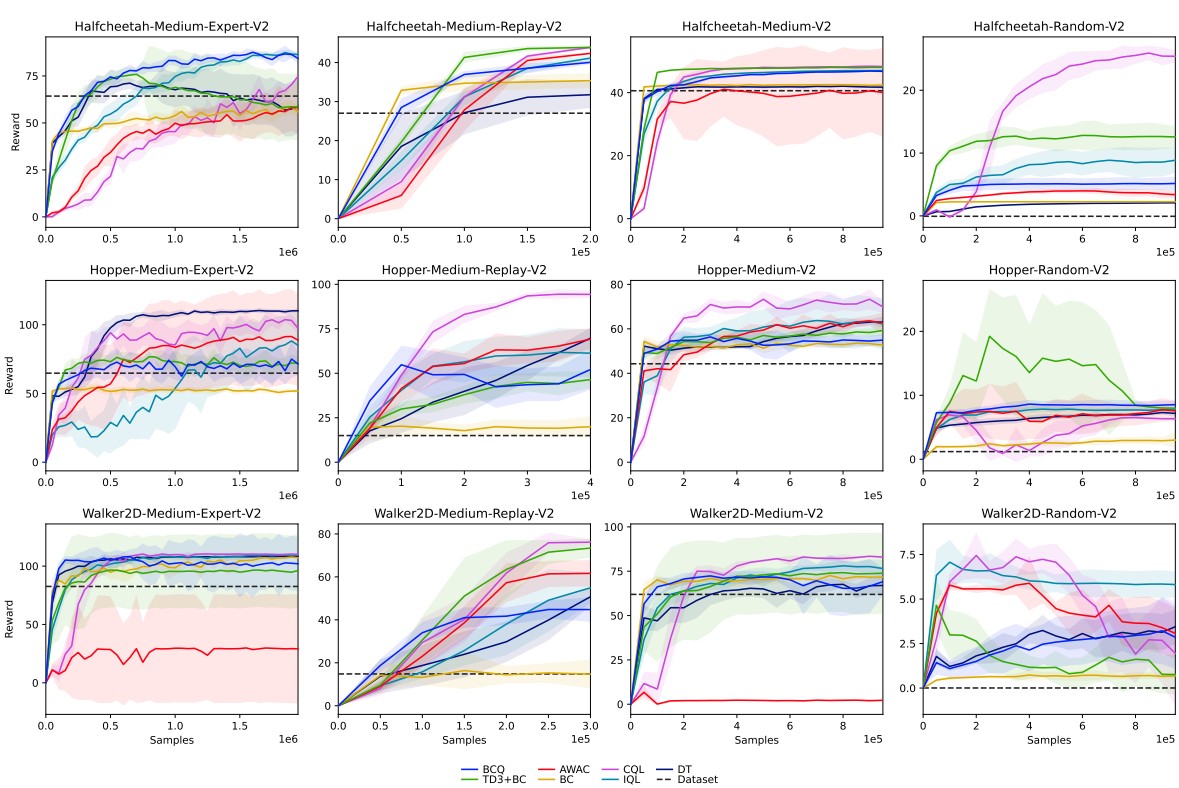

Figure 11: Performance curves on the D4RL benchmark of offline RL algorithms as a function of data points seen. Shaded regions represent standard deviation across 5 seeds.

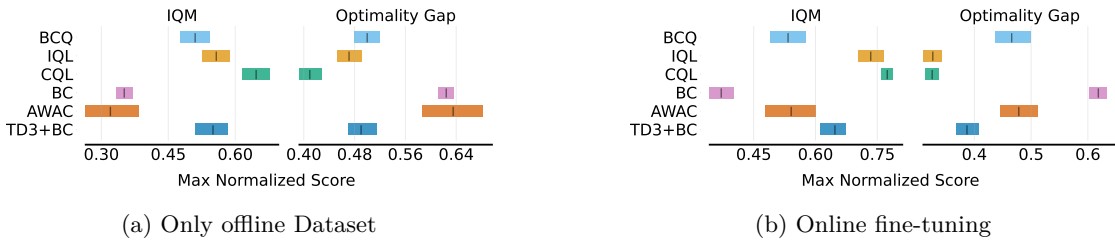

Figure 12: Performance curves for online fine-tuning. Each algorithm is given 500k steps in the simulator after sequential evaluation of the offline dataset. Dotted line indicates where online fine-tuning begins. Shaded regions represent standard deviation across 3 seeds.

(a) Only offline Dataset

(b) Online fine-tuning

Figure 13: Performance aggregated across environments using rliable. For IQM higher is better, while for Optimality gap, lower is better

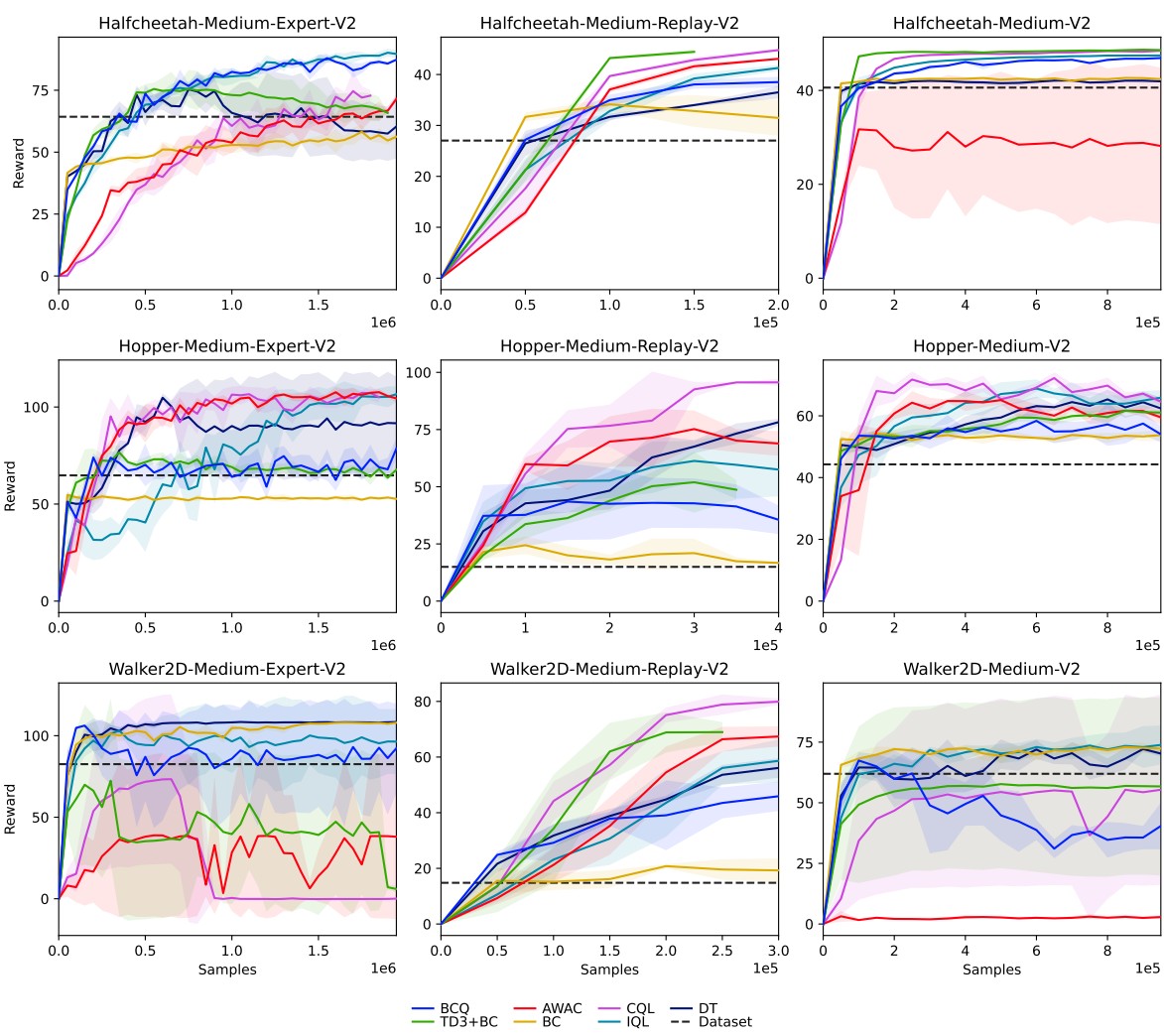

Figure 14: Performance curves on the D4RL benchmark with $K$ increased to 2. Shaded regions represent standard deviation across 3 seeds.

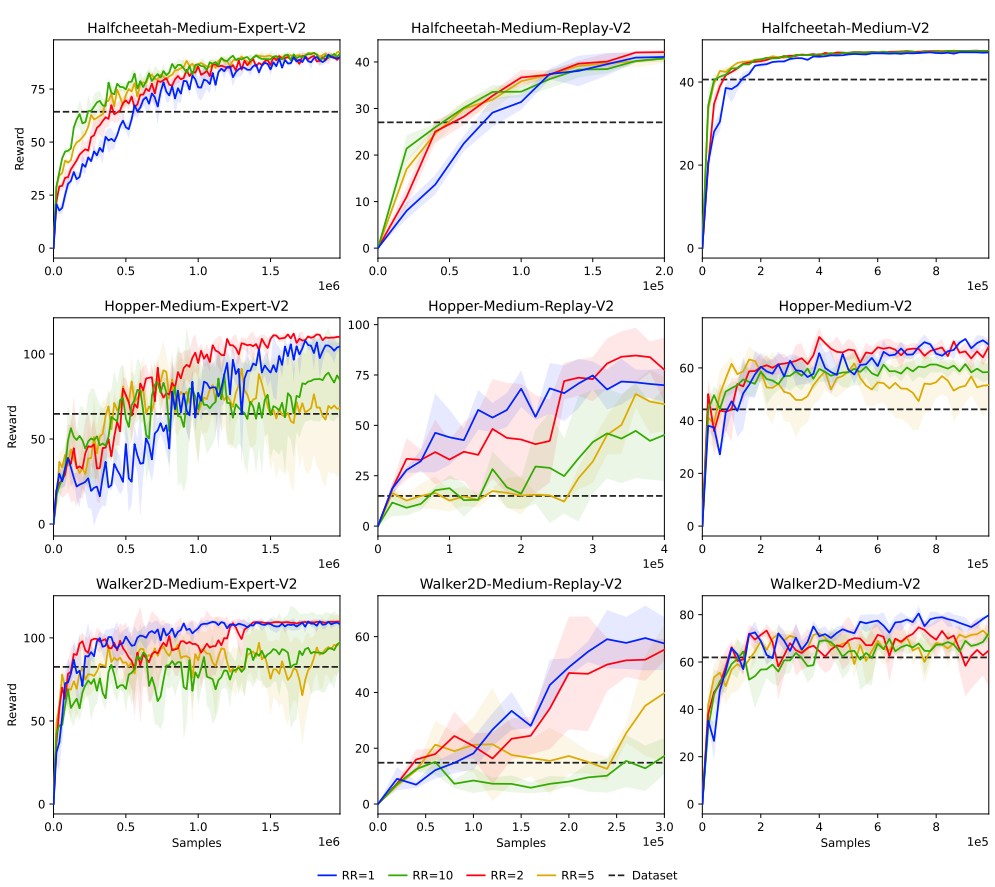

Figure 15: Comparison of replay ratios greater than 1.

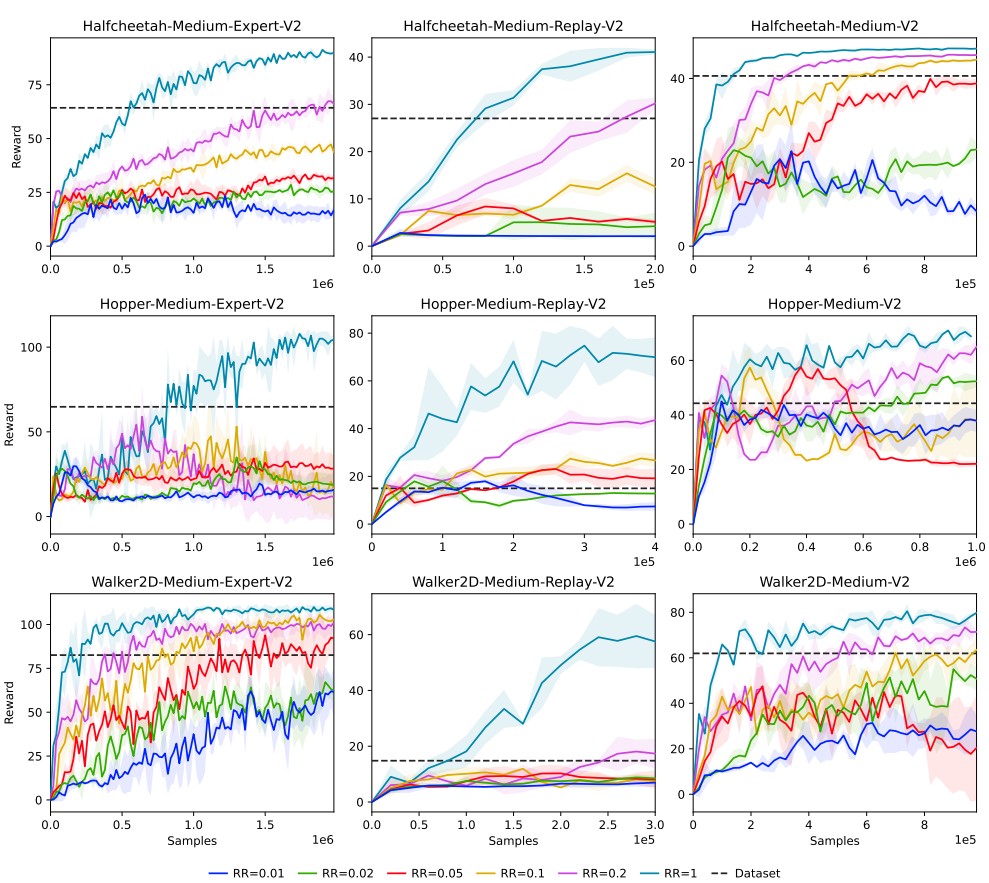

Figure 16: Comparison of replay ratios less than 1.

