# OpenReview forum: "Bridging the Gap Between Offline and Online Reinforcement Learning Evaluation Methodologies"
_TMLR — Accepted by TMLR_

### Review · Reviewer_wRve · 2023-08-14

**Summary Of Contributions:**

This paper introduces a method for evaluating offline RL algorithms, referred to as SeqEval. This approach assesses the algorithm's performance as a function of the available data, aiming to address limitations in past evaluations of offline RL methods. With this method, it becomes possible to investigate the sample efficiency and robustness of offline RL algorithms, while also enabling comparisons with online RL algorithms.

**Audience:**

Yes

**Broader Impact Concerns:**

None.

**Claims And Evidence:**

Yes

**Requested Changes:**

See weakness part.

**Strengths And Weaknesses:**

Strengths
1. The issue investigated in this article holds significant importance. Currently, the evaluation of most offline RL algorithms necessitates interaction with the environment, which contradicts the fundamental premise of offline RL. This aspect indeed calls for improvement.
2. The approach is generally applicable and easy to implement.

Weaknesses
1. The proposed method, which sequentially adds the existing offline data to the agent's replay buffer and periodically evaluates agent performance as a function of the available data, still requires interaction with the real environment. It does not fundamentally resolve the current issues in evaluating offline RL algorithms.
2. There is another paper that proposes an enhancement to evaluating offline RL algorithms [1]. That paper estimates policy performance by constraining the number of online evaluation iterations, thereby reducing the frequency of interactions with the online environment to some extent. That approach appears to be more aligned with practical applications. I would appreciate the author's insights on that paper and their perspective on its potential contributions to the field.
3. The evaluation method proposed in this article introduces additional parameters (gamma and K), which may vary for different offline RL algorithms. These parameters require extra time for tuning and introduce an additional burden.


[1] Kurenkov V, Kolesnikov S. Showing your offline reinforcement learning work: Online evaluation budget matters[C]//International Conference on Machine Learning. PMLR, 2022: 11729-11752.

---

### Review · Reviewer_ZR7g · 2023-08-21

**Summary Of Contributions:**

This paper presents a new evaluation methodology for offline RL, which measures the performance of algorithms in terms of data used in a similar manner to online RL, instead of measuring the performance after some arbitrarily set epochs. This is easy-to-implement because existing offline RL methods implements a replay buffer with fixed logged dataset and what the method should do is to use part of the buffer instead of using all of them. The proposed metric is investigated in D4RL benchmark and also the custom-made DMC benchmark and provide a new insight into the data-efficiency of existing offline RL algorithms.

**Audience:**

Yes

**Claims And Evidence:**

Yes

**Requested Changes:**

- It could be nice to provide a further detailed discussion on how to shuffle the fixed dataset for every run. Moreover, a kind of sensitivity analysis across each runs could be a nice addition.
- Providing metrics on more datasets (e.g., other datasets on D4RL, pixel-based dataset) could make it much more easier to use the metrics proposed in the paper. And this could make the benefit of using the proposed metric more persuasive.
- I think more investigation using the off-policy evaluation metrics and empirically showing its benefits for analyzing the performance of offline RL methods is crucial, considering the weakness W3. I noticed that authors also mentioned that `For example, if there are environments where a simulator is not available for periodic cheap evaluation, off-policy evaluation metrics Thomas et al. (2015); Wang et al. (2020) can be used.`. I think addressing this point would be very crucial for me to recommend acceptance of this paper.

**Strengths And Weaknesses:**

Strengths
- It's a new approach for measuring the performance of offline RL that also considers the data-efficiency, which is a bit overlooked point of view in offline RL.
- The main idea is very simple and clearly motivated.
- Writing is clear and easy to understand the main idea.

Weaknesses
- **W1:** It seems like how to choose the part of the buffer is very critical design choice, but this is not thoroughly investigated.
- **W2:** As a paper that introduces a new metric, there is a limited evaluation on few datasets.
- **W3:** Considering that the promise of offline RL is to learn the behavior using the fixed logged dataset, it's not clear how the proposed metric can be used for more practical scenarios where evaluation is prohibitively costly.

---

### Review · Reviewer_krMq · 2023-08-24

**Summary Of Contributions:**

This work propose a new approach of evaluation called Sequential Evaluation (SeqEval) for offline RL. SeqEval is a sequential approach to evaluate offline RL methods as a function of the training set size, rather than using the final performance trained on the whole offline dataset.

SeqEval is useful to evaluate the data efficiency and the robustness of algorithms to distribution shifts. Moreover, it bridges the gap between offline and online evaluation protocols.

In the experiments, SeqEval is adopted for the evaluation of representative offline algorithms, e.g., CQL, DT, TD3+BC, BCQ and etc, on both a standard setting and a distribution-shift setting with D4RL and DMC environments.

**Audience:**

Yes

**Claims And Evidence:**

Yes

**Requested Changes:**

1. I am curious about how different choices of $\gamma$ and $K$ influence the evaluation results. Would there be different ranks obtained from different choices of $\gamma$ and $K$. It seems that only $K=1,2$ and $\gamma=1$ are considered in this paper. Personally, I think this is quite limited. I would like to see some more choices maybe like $\gamma \in \\{1, 10, 50, 100\\} $ and $K \in \\{1, 2, 5, 10\\}$. I noticed the authors mentioned “we found that setting $\gamma$ and $K$ to 1 worked well in all datasets tested”, what is meaning of “worked well”?
2. Why are there some offline RL algorithms missing in specific plots, e.g., CQL is missing in Figure 3 and Figure 5(b), DT is missing in Figure 11?
3. I would like to know whether the samples in a medium-expert dataset are maintained in order like first the medium data then the expert data or fully mixed in random. Can I consider that even in the standard setting, SeqEval on the medium-replay and the medium-expert (if samples are in order like mentioned before) datasets contains distribution changes?

**Strengths And Weaknesses:**

### Strengths:

- The paper is well written and almost clear.
- The motivation makes sense to me and I believe SeqEval will be a very useful evaluation approach to offline RL community.
- Experiments are diverse with different settings (standard, distribution shift, online finetuning) and multiple metrics. The experimental results showcase the abilities of well-known offline RL algorithms from the angles of data efficiency and robustness.

&nbsp;

### Weaknesses:

- Although SeqEval is easy to implement, the hyperparameters $\gamma$ and $K$ (maybe $T_{0}$ also) introduced are not well investigated and discussed in this work. I am also aware that this may be something to be standardized or customized in the future.
- Some experimental details are missing. Please see my Requested Changes below.
- Some offline RL algorithms are missing in some plots without explanation.

---

### Review · Reviewer_gGYN · 2023-08-24

**Summary Of Contributions:**

This paper presents SeqEval, an experimental methodology for offline RL (+ finetuning) that attempts to provide a way to measure an algorithm’s dependency on dataset size and data distributional shift.

SeqEval is demonstrated by comparing a few recent baselines (CQL, DT, TD3+BC, …) on a subset of DMC and D4RL environments.

**Audience:**

Yes

**Broader Impact Concerns:**

None.

**Claims And Evidence:**

No

**Requested Changes:**

C1: I think it is necessary to compare SeqEval against the evaluation setups used in the baselines (a subset is fine). The goal is to figure out how SeqEval improves the understanding of each algo as presented by their own manuscripts. It would also highlight negative trade-offs.

C2: adding a sound justification for why the fine tuning bit is a good requirement to spec for (this could be done e.g. by lit review to see how many of the baselines also had finetuning experiments).

C3: SeqEval’s parameterization needs to be justified both empirically as well as philosophically to justifiy the significant addition in complexity. What is the objective of its two parameters? How would researchers choose them, or even compare between two instances of SeqEval?

**Strengths And Weaknesses:**

## Strengths

S1: the paper is tackling an unsexy issue whose resolution has massive potential impact on the field. Kudos!

S2: it’s a straightforward reutilisation of relatively simple and exhausting evaluation methods in RL, and thus easy for researchers to use.

## Weaknesses

W1: many of the papers cited in the lit review utilise a form of sequential eval where an offline policy is periodically evaluated out-of-training-dataset while training. This feels like very similar to the main contribution of the manuscript.

W2: “distributional shift” seems like a poor way of describing the variance in data / policy space for an arbitrary dataset. The manuscript seems to assume some notion of underlying quality for an arbitrary rollout (or SARS tuple?), which gets used as some measure of change in distribution. However, that’s really quite arbitrary, given how dependent this is on how difficult the task is, what are the data (and MDP) dynamics, the dimensionality of learning policy, etc. — Ultimately, this doesn’t seem like a useful “feature” to target for a general evaluation method, unless it can really demonstrate to be useful to gain consistent insights across a variety of (env, algorithm, model) sets.

W3: the method uses two params, which seem fairly important to understand the experimental dynamics. However they are immediately fixed in all experiments, and they are not explored at all. The manuscript doesn’t seem to discuss what this particular parameterization would bring to the table.

W4: the fine tuning part seems out of scope wrt. standardising offline RL eval. It’s not clear this is a strength of the method, since one could very well imagine using different algorithms with different eval requirements.

W5: the model card criteria don’t seem to control for different kinds of dataset sampling (e.g. uniform vs online prioritised vs pre-prioritised). Given that it’s a major source of complexity in offline RL algos, I’m not sure it is reasonable not to model it explicitly.

W6: I think the rationalizations of performance of the various tested algorithms are too bold. E.g. “this highlights that most of the tested algorithms are not very data-hungry” is hard to defend, given that it’s likely a function of the task + model dimensionality + algo being used. Other statements in the manuscript have similar issues.

---

### Decision · Action_Editors · 2023-10-09

**Recommendation:** Accept with minor revision

**Comment:**

All reviewers agreed that the paper tackles an important task. The main critique focused on the lack of a thorough investigation due to a limited set of experiments.
After the clarifications and additional results provided by the authors, the majority agreed that it is a paper that should be accepted.

This requires the addition of a detailed discussion of the new results, and I therefore recommend _Accept with minor revision_.

The authors should
- include their clarifications and comments made by the reviewers on the theoretical basis of the model
- include and discuss the additional results they provide, especially the role of $K$ and $\gamma$.

**Audience:**

The paper will be of interest to readers working on offline reinforcement learning methods.

**Claims And Evidence:**

The authors evaluate their proposed evaluation metric on a sequence of experimental tasks. An initial lack of experiments (e.g., an evaluation non the role of $\gamma$ and $K$) that was highlighted by the reviewers has been resolved.